# Machines and Mathematical Mutations: Using GNNs to Characterize Quiver Mutation Classes

**Jesse He**[†,⋆]**, Helen Jenne**[‡]**, Herman Chau**[†]**, Davis Brown**[‡]**,**
**Mark Raugas**[‡]**, Sara Billey**[†]**, Henry Kvinge**[‡,†]

[‡] Pacific Northwest National Laboratory
[⋆] University of California San Diego
[†] University of Washington
jeh020@ucsd.edu, henry.kvinge@pnnl.gov

## Abstract

Machine learning is becoming an increasingly valuable tool in mathematics, enabling one to identify subtle patterns across collections of examples so vast that they would be impossible for a single researcher to feasibly review and analyze. In this work, we use graph neural networks to investigate *quiver mutation*—an operation that transforms one quiver (or directed multigraph) into another—which is central to the theory of cluster algebras with deep connections to geometry, topology, and physics. In the study of cluster algebras, the question of *mutation equivalence* is of fundamental concern: given two quivers, can one efficiently determine if one quiver can be transformed into the other through a sequence of mutations? Currently, this question has only been resolved in specific cases. In this paper, we use graph neural networks and AI explainability techniques to discover mutation equivalence criteria for the previously unknown case of quivers of type $\tilde{D}_n$. Along the way, we also show that even without explicit training to do so, our model captures structure within its hidden representation that allows us to reconstruct known criteria from type $D_n$, adding to the growing evidence that modern machine learning models are capable of learning abstract and general rules from mathematical data.

## 1 Introduction

Examples play a fundamental role in the mathematical research workflow. Exploration of a large number of examples builds intuition, supports or disproves conjectures, and points towards patterns that are later formalized as theorems. While computer-aided calculations have long played an important role in mathematics research, modern machine learning tools (e.g., deep neural networks) have only recently begun to be more broadly applied. In this work, we show that a graph neural network trained to classify quivers into mutation equivalence classes learns representations that align with known mathematical theory. Where theory is unknown, we demonstrate how the model's representations can guide the discovery and proof of new mathematics.

Introduced by Fomin and Zelevinsky in [19], *quiver mutation* is a combinatorial operation on quivers (directed multigraphs) that arises from the notion of a *cluster algebra*. Identifying whether two quivers are *mutation equivalent* is generally a hard problem [29]. In some cases, such as type $A$ or type $D$ quivers, there exist results [5, 31] that characterize mutation equivalence classes in terms of structural conditions on the quiver. We train a graph neural network (GNN) to accurately classify quivers into one of six different mutation equivalence classes: types $A$, $D$, $E$, $\tilde{A}$, $\tilde{D}$, and $\tilde{E}$. Through a careful application of explainability tools and exploration of hidden activations, we find that the GNN extracts features from type $D$ quivers that align with the known characterization of [31]. Pushing this

38th Conference on Neural Information Processing Systems (NeurIPS 2024).

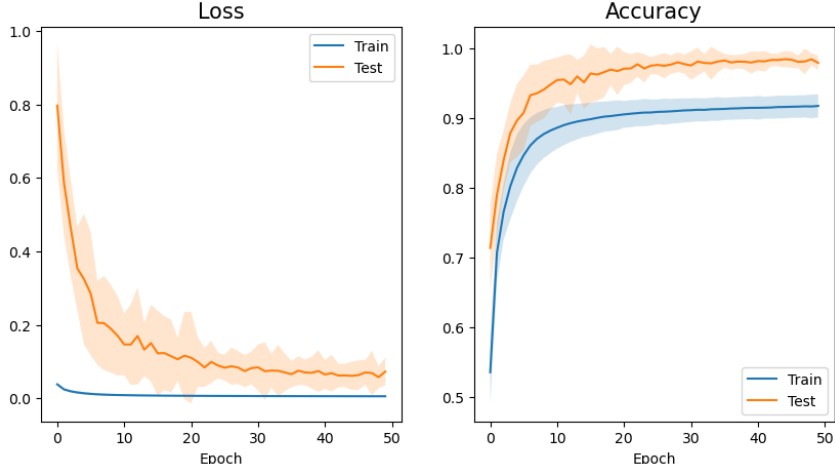

Figure 1: Average cross-entropy loss (left) and classification accuracy (right) on train and test sets across 10 trials of training. Testing accuracy is consistently higher than training accuracy, perhaps due to the absence of class $\tilde{E}$ in the test set and the fact that $\tilde{E}_8 = E_9$ in the train set.

further, we are able to use the same analysis of hidden activations to prove an explicit characterization of type $\tilde{D}$ quivers (Theorem 3.1), which was not known previously.

## 2 Quivers and quiver mutation

In their work on cluster algebras [19], Fomin and Zelevinsky introduce the notion of *matrix mutation* on skew-symmetric integer matrices. Skew-symmetric matrices and matrix mutations can be interpreted combinatorially as *quivers* and *quiver mutations*, which are the central objects of our study. In this section, we summarize quivers and quiver mutations. For further background, see Appendix A.1.

**Definition 2.1.** A *quiver Q* is a directed multigraph with no loops or 2-cycles. The number of parallel edges between a pair of vertices is represented by a positive integer weight.

**Definition 2.2.** The *mutation* of a quiver $Q$ at a vertex $j$ is the quiver $\mu_j(Q)$ obtained by performing the following: (i) For each path $i \to j \to k$ in $Q$, add an edge $i \to k$. (ii) Reverse all edges incident to $j$. (iii) Remove all 2-cycles created from the previous two steps.

Two quivers $Q$ and $Q'$ are *mutation equivalent* if $Q'$ can be obtained from $Q$ by a sequence of mutations. For any vertex $j$ in a quiver $Q$, mutation at $j$ is an involution: $\mu_j(\mu_j(Q)) = Q$. The set of quivers mutation equivalent to a quiver $Q$ is called the *mutation class* of $Q$.

We consider the mutation classes of quivers that are of simply laced Dynkin type or affine Dynkin type. These quivers have no parallel edges, and their underlying undirected graphs are shown in Fig. 7 in Appendix A. Quivers of type $D$ and type $\tilde{D}$ are the focus of our explainability analysis, but all the quivers shown in Fig. 7 are included in our training and test sets.

**The machine learning task:** Train a classifier $\Phi$ to predict the mutation class of a quiver of type $A$, $D$, $E$, $\tilde{A}$, $\tilde{D}$, or $\tilde{E}$. Because prior work has characterized quiver mutation classes based on the presence of particular subgraphs, we adapt the most expressive GNN architecture for recognizing subgraphs to support directed edges with edge attributes [22, 33]. We describe our network $\Phi$ as a 4-layer **Dir**ected **G**raph **I**somorphism **N**etwork with **E**dge features (DirGINE). Each hidden layer has width 32, and the output layer has width 6, since there are 6 classes. We will train $\Phi$ on quivers with 6, 7, 8, 9, and 10 nodes, and test on quivers with 11 nodes. Our model achieves high accuracy (99.2%) on the test set (Fig. 1). More details are provided in Appendix B.1.

# 3 Extracting characterizations of mutation classes

In this section, we present our main result: a characterization of the mutation class of type $\tilde{D}_{n-1}$ quivers, obtained by probing our trained GNN. Characterizations of the mutation class of type $A_n$ and type $D_n$ quivers are known [5, 31], but to our knowledge, the type $\tilde{D}_{n-1}$ case was previously unknown.

**Theorem 3.1.** *The mutation class of class $\tilde{D}_{n-1}$ quivers is $\mathcal{M}_{n-1}^{\tilde{D}}$, the collection of quivers of paired types together with Types V, Va, Vb, V', Va', Vb', VI, and VI' (as described in Appendix D).*

The discovery of the precise types in Theorem 3.1 was aided by insights from edge attributions and latent space clustering in our GNN. The proof that the characterization in Theorem 3.1 is complete follows from a similar argument used by Vatne [31] to decompose quivers of type $D_n$.

## 3.1 Recovering a known characterization

We developed and validated our explainability techniques by recovering the known characterization of quivers of type $A_n$ and type $D_n$. The type $A_n$ quivers consist of all quivers that are mutation equivalent to the quiver (6) in Appendix C. Buan and Vatne gave a combinatorial characterization of all quivers in this mutation class, which we refer to by $\mathcal{M}_n^A$.

**Theorem 3.2** (Buan and Vatne [5]). *A quiver is in $\mathcal{M}_n^A$ if and only if: (i) All cycles are oriented 3-cycles. (ii) Every vertex has degree at most four. (iii) If a vertex has degree four, two of its edges belong to the same 3-cycle, and the other two belong to a different 3-cycle. (iv) If a vertex has degree three, two of its edges belong to a 3-cycle, and the third edge does not belong to any 3-cycle.*

Vatne's classification [31] of the mutation class $\mathcal{M}_n^D$ of type $D_n$ quivers builds upon the quivers in $\mathcal{M}_n^A$. Each quiver in $\mathcal{M}_n^D$ decomposes into a collection of subquivers joined by gluing certain vertices known as *connecting vertices*. A vertex $c$ is a connecting vertex if $c$ is either degree one, or if $c$ is degree two and part of an oriented 3-cycle.

**Theorem 3.3** (Vatne [31]). *The quivers in $\mathcal{M}_n^D$ are divided into four subtypes shown in Fig. 2, where $\Gamma$, $\Gamma'$, and $\Gamma''$ denote subquivers that are in mutation class $\mathcal{M}_k^A$ for some integer $k$.*

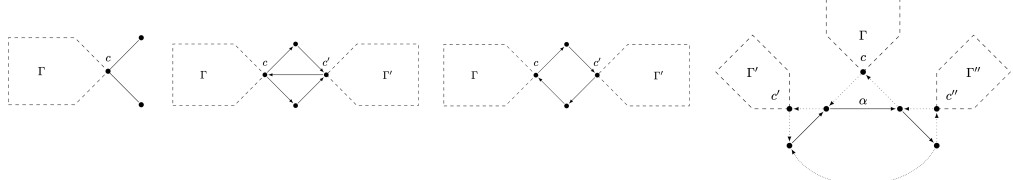

Figure 2: Types I, II, III, and IV in $\mathcal{M}_n^D$, from left to right. The subquivers $\Gamma$, $\Gamma'$, and $\Gamma''$ are type $A$, and $c$, $c'$, and $c''$ are connecting vertices. Unoriented edges may be oriented in either direction.

Since GNNs are capable of recognizing subgraphs and structural patterns as expressively as the classical Weisfeiler-Lehman graph isomorphism test [33, 34], we conjectured that our performant GNN model captured the same subtype motifs identified by human mathematicians. Using the explanation method PGExplainer described in Appendix B.3, we investigated type $D_n$ quivers in relation to Vatne's characterization. In Fig. 3, darker edges are more important for predicting type $D_n$ quivers, while lighter edges are less important. Subquivers of types I, II, III, and IV in Vatne's characterization are given high attribution.

Fig. 3 strongly suggests that our GNN recognizes the same subtypes as in Vatne's characterization. However, one should be careful in this interpretation, as there is substantial literature showing that it is easy to misinterpret post-hoc explainability methods [21, 23]. Thus, we also examine the embeddings of type $D_n$ quivers in the model's latent space. We use principal component analysis (PCA) to reduce the dimension of the embedding from the model width of 32 to 2 dimensions for visualization. The resulting graph embeddings, plotted in Fig. 4, show a clear separation of the different subtypes. In fact, the layer 3 embeddings in the original 32-dimensional embedding space can be separated by a linear classifier with $99.7 \pm 0.0\%$ accuracy. Subtypes I through IV are not labeled in the training data, so this analysis provides strong evidence that a GNN is capable of re-discovering the same abstract,

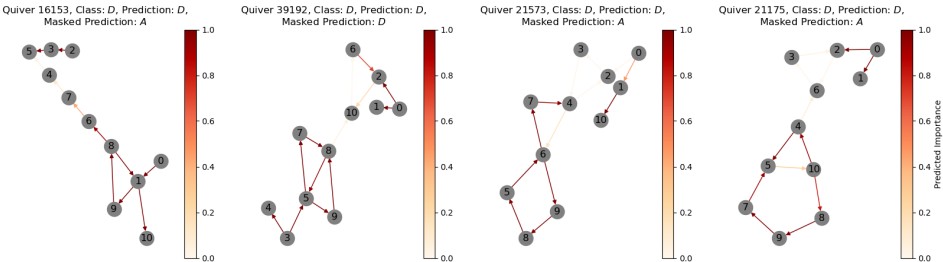

Figure 3: Edge attributions from PGExplainer on type $D_{11}$ quivers of each subtype (from left to right, Type I, Type II, Type III, Type IV). The masked prediction is the GNN prediction when highly attributed edges are removed.

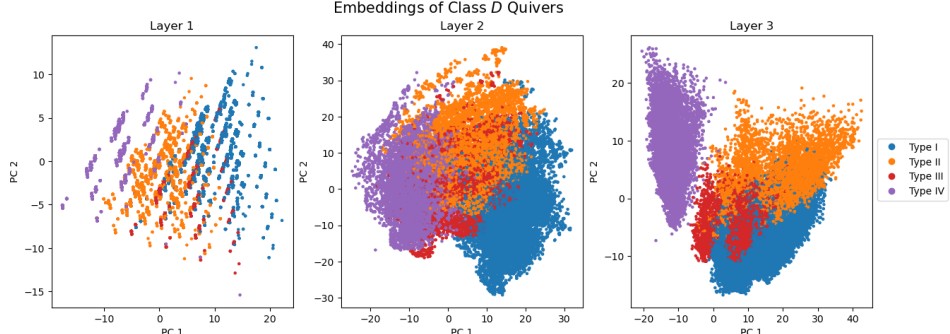

Figure 4: PCA of latent space embeddings for mutation class $D$ quivers colored by subtypes.

general characterization rules that align with known theory. Furthermore, explainability methods such as PGExplainer can be leveraged to extract these rules from the model.

## 3.2 Additional experiment: label-flipping

To see how the model is using the type $D$-specific subquivers from Theorem 3.3, we examine the model's predictions when the vertices identified by PGExplainer are removed. If the model is primarily keying into the type $D$ motif, removing this should result in a quiver of type $A$.

We find that across all $32,066$ test examples from type $D$, a plurality (14,916 or 46.5%) of the predictions flip to $A$, as we would expect if it was using the characterization from Theorem 3.3. Of the remaining examples, most (14,238 or 44.4% of the total) flip to a predicted class of $E$, with the next-largest being $D$ (no flip) at 2,581 or 8.0%. Finally, 264 quivers (0.08%) flip to $\tilde{A}$, while 67 quivers (0.02%) flip to $\tilde{D}$. None of the predictions flip to $\tilde{E}$. We believe the large number of flips to $E$ is due to the PGExplainer attributions for type $D$ being inexact, perhaps because PGExplainer generalizes imperfectly or because some edges may not contribute positively to $D$ but rather contribute negatively to other classes. As a result, many of the quivers where we remove highly attributed edges may be out-of-distribution for the model. Since type $E$ is the only class which contains mutation-infinite quivers, it is perhaps not surprising that the model would predict these out-of-distribution quivers are of type $E$.

## 3.3 Discovering a new characterization

This explainability workflow is then applied to type $\tilde{D}_{n-1}$, where no known characterization existed. Compared to type $D_n$, the mutation class of $\tilde{D}_{n-1}$ quivers contains many more diverse subtypes and combinations of motifs. Careful analysis of edge attributions in the $\tilde{D}_{n-1}$ led to our discovery of paired types along with novel subtypes V, Va, Vb, V', Va', Vb', VI, VI'. Based on our strategy in Section 3, we plot PCA reductions of the latent space in Fig. 5. We can see that the quivers that do not correspond to paired subtypes, colored as "Other", separate clearly into two clusters in layer

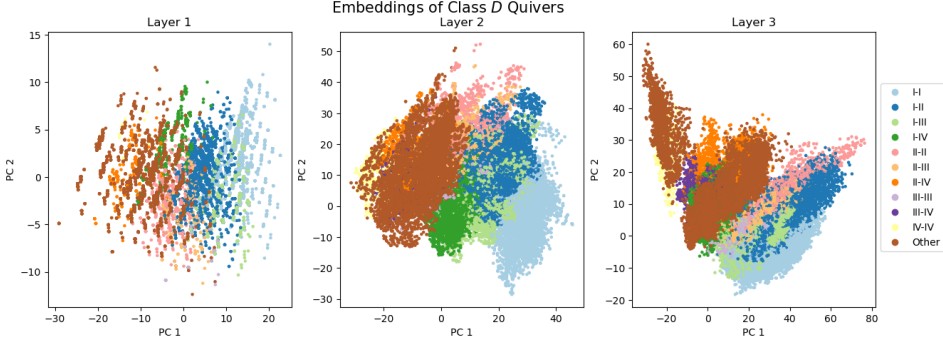

Figure 5: PCA reductions of latent space embeddings for mutation class $\tilde{D}_{10}$ quivers colored by what we call *paired types* and *"Other"* (see Appendix D).

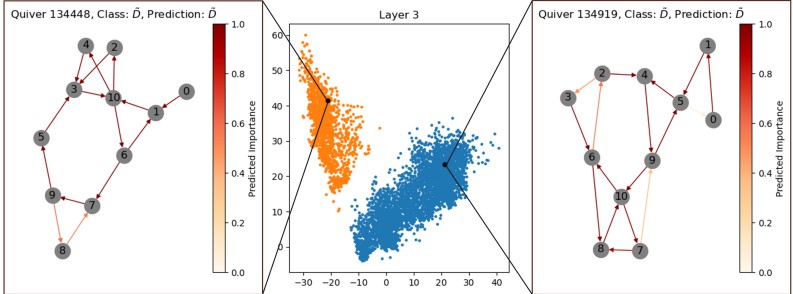

Figure 6: PCA of clustered layer 3 latent space embeddings for quivers in $\mathcal{M}_{10}^{\tilde{D}}$ not of paired type (middle), with selected examples from each cluster (left, right). Edges are colored by PGExplainer attributions. The quiver on the left is of Type V while the quiver on the right is of Type VI.

3. By isolating these quivers and performing $k$-means clustering with $k = 2$, the model guides our characterization of the remaining class $\tilde{D}$ subtypes. Fig. 6 shows examples from each cluster. (More examples are given in Appendix E.) See Appendix D for further details of the discovery process and Appendix D.4 for a rigorous proof of Theorem 3.1.

## 4   Conclusion

In this work, we analyzed a graph neural network trained to classify quivers into one of 6 different mutation equivalence classes. Using explainability techniques, we provided evidence that the model learns prediction rules that align with existing theory for type $D_n$ quivers. Moreover, our result emerged from the model in an unsupervised manner—the model is not given any subtype labels, and yet is able to identify relevant subquivers that characterize type $D_n$ quivers. Applying the same explainability techniques to an unknown case, we discovered and proved a characterization of the mutation class of $\tilde{D}_{n-1}$ quivers, a case which had not previously been described in this manner. Our work contributes to the growing evidence that machine learning can be a valuable tool in the mathematician's workflow by identifying novel patterns in mathematical data.

## Acknowledgments and Disclosure of Funding

The authors would like to thank Scott Neville and Kayla Wright for helpful discussions. This research was supported by the Mathematics for Artificial Reasoning in Science (MARS) initiative via the Laboratory Directed Research and Development (LDRD) investments at Pacific Northwest National Laboratory (PNNL). PNNL is a multi-program national laboratory operated for the U.S. Department of Energy (DOE) by Battelle Memorial Institute under Contract No. DE-AC05-76RL0-1830.

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

# A  Additional background

## A.1  Cluster algebras

Quivers and quiver mutations are central in the combinatorial study of *cluster algebras*, which is a relatively new but very active research area, with connections to diverse areas of mathematics. This section provides a very high-level description of how the quiver mutation problem fits within the broader context of cluster algebras.

A cluster algebra is a special type of commutative ring that is generated (in the algebraic sense) via a (possibly infinite) set of generators that are grouped into *clusters*. A cluster algebra may have finitely or infinitely many generators, but the size of each cluster is always finite and fixed. A cluster algebra is said to be of *rank $n$* if each of the clusters contains $n$ generators, called *cluster variables*. These clusters are related via an *exchange property* which tells us how to transform one cluster to another [19]. It turns out that there is a nice combinatorial interpretation of this transformation when we interpret clusters as quivers with each generator corresponding to a vertex in the quiver. Then quiver mutation describes this exchange of cluster variables. In this setting, the mutation equivalence problem asks when two clusters generate the same cluster algebra.

In [20], Fomin and Zelevinsky gave a complete classification of cluster algebras for which there are a finite number of cluster variables. Such algebras are called *finite type*. Amazingly, they correspond exactly to the Cartan-Killing classification of semisimple Lie algebras. Their result says that a quiver associated to a cluster algebra of finite type must be mutation equivalent to an orientation of a Dynkin diagram. However, this result does not give an algorithm for checking whether a quiver is of finite type. To answer this question, Seven [28] gave a full description of the associated quivers by computing all minimal quivers of infinite type. Since then, several other researchers have provided explicit characterizations of particular mutation classes of quivers [4, 5, 31]. Our main result follows these: we give an explicit characterization of quivers of type $\tilde{D}_n$, akin to the characterization of quivers of type $D_n$ given in [31].

**Remark A.1.** Some authors designate some vertices in a quiver to be *mutable* (that is, eligible for mutation), and *frozen* otherwise (corresponding to frozen variables in a cluster). Frozen vertices may not be mutated at, nor may any incident arrows be created, deleted, or reversed. However, in this paper, all vertices are always mutable.

**Definition A.2.** We say a quiver $Q$ is *mutation-finite* if its mutation class $[Q]$ is finite, and *mutation-infinite* otherwise.

**Definition A.3.** Given a starting quiver $Q$, the *mutation depth* of a quiver $Q' \in [Q]$ (with respect to $Q$) is the minimum number of mutations required to obtain $Q'$ from $Q$.

Quivers of type $\tilde{D}_n$ are *mutation-finite*, meaning they have a finite mutation equivalence class. Mutation-finite quivers and their associated cluster algebras are of interest to many cluster algebraists. Felikson, Shapiro, and Tumarkin [14] gave a description of the mutation-finite quivers in terms of *geometric type* (those arising from triangulations of bordered surfaces), the $E_6, E_7, E_8$ Dynkin diagrams and their extensions, and two additional exceptional types $X_6$ and $X_7$ identified by Derksen and Owen [13]. Specifically, they showed that mutation-finite quivers must either be decomposable into certain *blocks* or contain a subquiver which is mutation equivalent to $E_6$ or $X_6$. It is worth noting that classification in the mutation-finite setting has proven to be more challenging than in the finite setting. The classification of *mutation-finite* cluster algebras in the case with no frozen variables was achieved nearly a decade after Fomin and Zelevinsky classified finite cluster algebras [14, 15], and the general case was solved only last year [16].

## A.2  Mathematics and machine learning

Machine learning has recently gained traction as a tool for mathematical research. Mathematicians have leveraged its ability to, among other things, identify patterns in large datasets. These emerging applications have included some within the field of cluster algebras. For instance, in [7], Cheung et al. train machine learning models to classify semi-standard Young tableaux (SSYT) according to whether or not they correspond to a cluster variable in a Grassmannian cluster algebra, and if so, which cluster algebra the SSYT corresponds to. Based on the behavior of these models, Cheung et al. then pose a number of conjectures regarding SSYT and cluster algebras. Most similar to our own

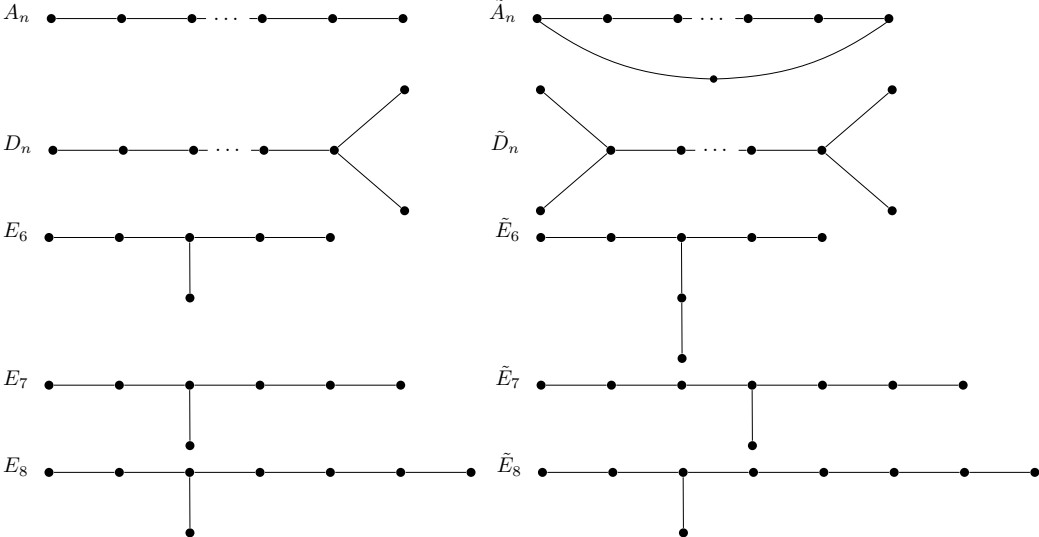

Figure 7: Simply laced Dynkin diagrams and their extensions.

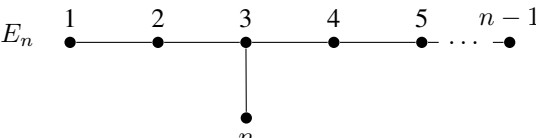

Figure 8: Coxeter-Dynkin diagram for $E_n$, $n \geq 6$. Quivers of type $E_n$ are only mutation-finite for $n = 6, 7, 8, 9$.

work, Bao et al. [3] and Dechant et al. [11] also study mutation using machine learning tools. Unlike our work, this research does not aim to establish new theorems around mutation equivalence classes, focusing rather on the performance of models on different versions of this problem. As such, their work does not apply any explainability methods to their models. Though unrelated to cluster algebras, Davies et al. [9] take an approach similar to the one taken here: using machine learning to guide mathematicians' intuition. They focus on two questions: one related to knot theory and one related to representation theory.

Due to the existence of unambiguous ground truth and known algorithmic solutions, there has also been renewed interest in using mathematical tasks to better analyze how machine learning models learn tasks at a mechanistic level, including the emergence of reasoning in large models. For example, in [8], the authors use group operations to investigate the question of *universality* in neural networks. Group multiplication is also used in [30] to investigate the *grokking* phenomenon. The idea of *mechanistic interpretability*—explaining model behavior by identifying the role of small collections of neurons—is also demonstrated in [36], where Zhong et al. are able to recover two distinct algorithms from networks trained to perform modular arithmetic, and [25], where Liu et al. find evidence that a network trained to predict the product of two permutations learns group-theoretic structure.

## B  Model and training details

Graph neural networks (GNNs), introduced in [12, 24], are a class of neural networks which operate on graph-structured data via a *message-passing* scheme. Given an (attributed) graph $G = (V, E)$ with node features $x_v \in \mathbb{R}^p$ for each node $v \in V$ and $e_{uv} \in \mathbb{R}^q$ for each edge $(u, v) \in E$, each layer of the network updates the node feature by aggregating the features of its neighbors. Because prior work has characterized quiver mutation classes based on the presence of particular subgraphs, we use the most expressive GNN architecture for recognizing subgraphs [33]. To this end, we adopt a version of the graph isomorphism network (GIN) introduced in [33] and modified in [22] to support

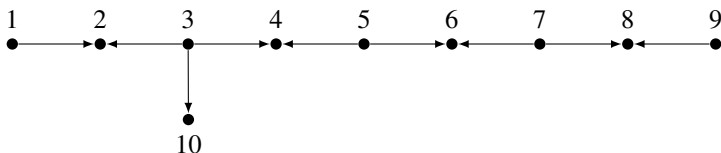

Figure 9: Default orientation of $E_{10}$ in Sage. Mutation depth is assessed with respect to this orientation for generating data in Sage.

edge features. Since quivers are directed graphs, we adopt a directed message-passing scheme with separate message-passing functions along each orientation of an edge. We refer to our architecture as a **Dir**ected **G**raph **I**somorphism **N**etwork with **E**dge features (DirGINE), and denote the network itself by $\Phi$. Formally, the $\ell$-th layer is given by

$$x_v^{(\ell)} = \text{ReLU}\left(W^{(\ell)}x_v^{(\ell-1)} + \sum_{(u,v)\in E} \varphi_{\text{in}}^{(\ell)}\left(x_u^{(\ell-1)}, e_{uv}\right) + \sum_{(v,w)\in E} \varphi_{\text{out}}^{(\ell)}\left(x_w^{(\ell-1)}, e_{vw}\right)\right) \quad (1)$$

where $W^{(\ell)}$ is an affine transformation and $\varphi_{\text{in}}^{(\ell)}$ and $\varphi_{\text{out}}^{(\ell)}$ are feedforward neural networks with 2 fully connected layers. Because we wish to classify graphs, we use *sum pooling*. That is, in the final layer $L$ we can assign a vector to the entire graph $G$ by adding the vectors associated with each vertex in the layer. We write

$$\Phi(G) = \Phi^{(L)}(G) = \sum_{v\in V(G)} x_v^{(L)}. \quad (2)$$

The expressive power of graph neural networks is intimately connected to the classical Weisfeiler-Lehman (WL) graph isomorphism test [32]. Given an undirected graph with constant node features and no edge features, a graph neural network cannot distinguish two graphs which are indistinguishable by the WL test [33], and graph neural networks are able to count some (but not all) substructures [6]. In our case, operating on directed graphs with edge features slightly enhances the expressive power of our network. As we saw in Section 3, the ability to distinguish substructures is crucial to their application in classifying quiver mutation classes.

## B.1 Model training

We train a 4-layer DirGINE GNN to classify quivers into six mutation classes: $A$, $D$, $E$, $\tilde{A}$, $\tilde{D}$, and $\tilde{E}$. Each hidden layer has width 32, and the output layer has width 6, since there are 6 classes. The training data is generated with Sage [27] and consists of

- All quivers of types $A$, $D$, $\tilde{A}$, and $\tilde{D}$ on 7, 8, 9, and 10 nodes.

- All quivers of type $\tilde{E}$. (Type $\tilde{E}$ is only defined for 7, 8, and 9 nodes, corresponding to extended versions of $E_6$, $E_7$, and $E_8$, respectively. All quivers of type $\tilde{E}$ are mutation-finite.)

- All quivers of type $E$ for $n = 6, 7, 8$. (The Dynkin diagram $E_9$ is the same as the extended diagram $\tilde{E}_8$.) Type $E$ is only mutation-finite for $n = 6, 7, 8$. and coincides with $\tilde{E}_8$ for $n = 9$.

- Quivers of type $E_{10}$ up to a mutation depth of 8, with respect to Sage's standard orientation for $E_{10}$ (Fig. 9). (While type $E$ is mutation finite for $n \leq 9$, $E_{10}$ is mutation-infinite).

The test set consists of quivers on 11 nodes. We use the full mutation classes of $A_{11}$, $\tilde{A}_{10}$, $D_{11}$ and $\tilde{D}_{10}$, and again generate quivers up to a mutation depth of 8 for $E_{11}$. The number of quivers of each size from each class can be found in Table 1 in Appendix B.2. Note that type $\tilde{E}$ is absent from the test set, because $\tilde{E}$ is not defined for 11 nodes.

We train with the Adam optimizer for 50 epochs with a batch size of 32 using cross-entropy loss with $L_1$ regularization ($\gamma = 5 \times 10^{-6}$) using an Nvidia RTX A2000 Laptop GPU. Fig. 1 shows the average cross-entropy loss and classification accuracy by epoch across 10 trials. We take the best epoch from training, achieving 99.2% accuracy on the test set.

Note that while the differences between the train and test set (particularly the absence of type $\tilde{E}$ from the test set) might be problematic if our goal was to simply assess whether a machine learning model can differentiate between quivers of different mutation types, our primary goal is to extract mathematical insights from the features the model learns for types $D$ and $\tilde{D}$. As such, for this work the test set was mostly used as a guide for when a model was sufficiently performant to justify the application of explainability tools. Ultimately, all our results are justified with mathematical proofs. As the test accuracy above suggests, this model learned to be highly accurate at classifying these particular mutation classes.

## B.2 Data generation

Quivers were generated using Sage [27]. For training and inference, each quiver was converted to PyTorch Geometric [17]. Following the representation convention in Sage, $k$ parallel edges are represented by a single edge with edge attribute $(k, -k)$, and each vertex is initialized with constant node feature. Table 1 shows the number of quivers of each class and size generated.

| | **Train** | | | | | |
|---|---|---|---|---|---|---|
| $n$ | $A_n$ | $D_n$ | $E_n$ | $\tilde{A}_{n-1}$ | $\tilde{D}_{n-1}$ | $\tilde{E}_{n-1}$ |
| 7 | 150 | 246 | 416 | 340 | 146 | 132 |
| 8 | 442 | 810 | 1,574 | 1,265 | 504 | 1,080 |
| 9 | 1,424 | 2,704 | — | 4,582 | 1,868 | 4,376 |
| 10 | 4,522 | 9,252 | 10,906 | 16,382 | 6,864 | — |
| | **Test** | | | | | |
| 11 | 14,924 | 32,066 | 24,060 | 63,260 | 25,810 | — |

Table 1: Number of quivers of each type and size in train and test sets.

## B.3 Explaining GNNs

In order to extract mathematical insight from a trained GNN model $\Phi$, we require a way to *explain* its predictions by identifying the substructures that are responsible for its predictions. That is, for each graph $G$, we wish to identify a small subgraph $G_S$ such that $\Phi(G) \approx \Phi(G_S)$. While a number of post-hoc explanation methods exist for GNNs, most fall into one of two categories:

(i) *Gradient-based* methods use the partial derivatives of the model output with respect to input features. A larger gradient is assumed to mean that a feature is more important.

(ii) *Perturbation-based* methods observe how the model's predictions change when features are removed or distorted. Larger changes indicate greater importance.

We use the GNN explanation method PGExplainer [26], a perturbation-based method which trains a neural network $g$ to identify important subgraphs. For the input graph $G$, the explanation network operates on each edge $(u, v)$. Using the final node embeddings of $u$ and $v$ as well as any edge features $e_{uv}$, $g$ produces an attribution

$$\omega_{uv} = g(x_u^{(L)}, x_v^{(L)}, e_{uv}). \tag{3}$$

Here $g$ is implemented as an MLP followed by a sigmoid function to ensure that $0 \leq \omega_{uv} \leq 1$. Then rather than produce a "hard" subgraph as our explanation, the attribution matrix $\Omega = (\omega_{ij})$ can be seen as a "soft" mask for the adjacency matrix $A(G)$. That is, instead of providing a binary 0-1 attribution for each edge, PGExplainer provides an attribution $\omega_{ij} \in [0, 1]$. We then use the weighted graph with adjacency matrix $\Omega \odot A(G)$ for $G_S$ (where $\Omega \odot A(G)$ is the elementwise product).

PGExplainer follows prior work [35] in interpreting $G_S$ as a random variable with expectation $\Omega = \mathbb{E}[A(G_S)]$, where each edge $(i, j)$ is assigned a Bernoulli random variable with expectation $\omega_{ij}$. PGExplainer then attempts to maximize the *mutual information* $I(\Phi(G), G_S)$. However, because this is intractable in practice, the actual optimization objective is

$$\min_{\Omega} \text{CE}(\Phi(G), \Phi(G_S)) + \alpha\|\Omega\|_1 + \beta H(\Omega). \tag{4}$$

Here $CE(\Phi(G), \Phi(G_S))$ is the cross-entropy loss between the predictions $\Phi(G)$ and $\Phi(G_S)$, $\|\Omega\|_1$ is the $L_1$-norm of $\Omega$,

$$H(\Omega) = - \sum_{(i,j) \in E} \sum_{(i,k) \in E} [(1 - \omega_{ij}) \log(1 - \omega_{ik}) + \omega_{ij} \log(\omega_{ik})], \tag{5}$$

and $\alpha$ and $\beta$ are hyperparameters. The $\|\Omega\|_1$ term acts as a size constraint, penalizing the size of the selected $G_S$. The $H(\Omega)$ term acts as a connectivity constraint, penalizing instances where two incident edges are given very different attributions. By training a neural network to compute $\Omega$, PGExplainer allows us to generate explanations for new graphs very quickly, as well as take a more global view of the model behavior.

While PGExplainer's effectiveness is mixed across different comparisons [1, 2], it has been shown to be effective at providing model-level substructure explanations for graph classification tasks. For example, when applied to a GNN trained on the MUTAG dataset [10] to predict the mutagenicity of molecules, PGExplainer is regularly able to identify that the model predicts mutagenicity based on the presence of nitro ($NO_2$) groups [26]. As we will see in Section 3, the ability of PGExplainer to identify explanatory graph motifs makes it suitable for our purposes. To analyze our trained DirGINE, we train PGExplainer's internal neural network on 1000 randomly selected instances from the training set for 5 epochs with hyperparameters $\alpha = 2.5$ and $\beta = 0.1$.

## C  Characterizing quiver mutation classes

In this section, we provide additional details around extracting the characterization of class $D_n$ quivers from our model. Appendix C.1 provides mathematical background used by [31] in Theorem 3.3. Section 3.2 details an additional experiment we performed to probe our GNN model.

### C.1  Preliminaries

We will use the following well-known lemma (see, e.g., [31]):

**Lemma C.1.** *If quivers $Q_1$ and $Q_2$ have the same underlying graph $T$ (that is, the graph obtained by forgetting the orientation of edges), where $T$ is a tree, then $Q_1$ and $Q_2$ are mutation equivalent.*

This lemma is helpful because it allows us to talk about the mutation classes of a simply laced Dynkin diagram, ignoring edge orientations. In particular, we can use (6) and (7) as starting orientations for types $A$ and $D$, respectively.

$$\tag{6}$$

$$\tag{7}$$

Our main classes of concern (types $A_n$, $D_n$, and $\tilde{D}_n$) are of *geometric type*, meaning they can be associated with triangulations of bordered, possibly punctured surfaces. Consequently, quivers in these classes can all be decomposed into collections of subgraphs known as *blocks* [15].

**Definition C.2.** A *block* is one of five graphs shown in Fig. 10, where each vertex is either an *outlet* or a *dead end* [18]. A connected quiver $Q$ is *block-decomposable* if it can be obtained by gluing together blocks at their outlets, such that each vertex is part of at most two blocks. Formally,

1. Take a partial matching of the combined set of outlets (no outlet may be matched to an outlet from the same block);

2. Identify the outlets in each pair of the matching;

3. If the resulting quiver contains a pair of edges which form a 2-cycle, remove them.

**Lemma C.3** (Vatne [31]). *Let $\Gamma \in \mathcal{M}_n^A$, $n \geq 2$, and let $c$ be a connecting vertex for $\Gamma$. Then there exists a sequence of mutations on $\Gamma$ such that: (i) $\mu_c$ does not appear in the sequence (that is, we do not mutate at $c$); (ii) The resulting quiver is isomorphic to (6); (iii) Under this isomorphism, $c$ is mapped to 1.*

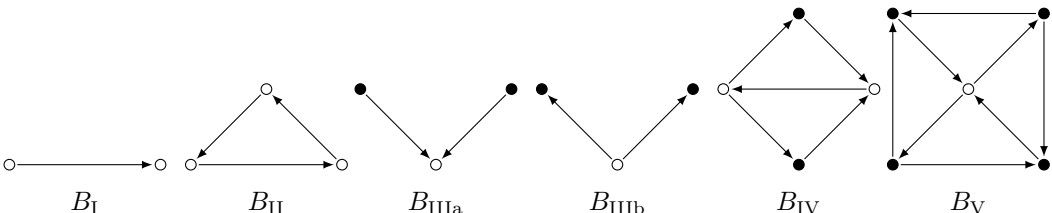

Figure 10: Blocks of type I-V introduced by [18]. Open circles denote *outlets*, which may be identified with at most one outlet from another block. Closed circles represent *dead ends*, which may not be identified with any other vertex.

# D  The mutation class of $\tilde{D}_n$ quivers

In this section, we use our trained model and explainability techniques to discover a characterization of the mutation class of $\tilde{D}_n$ quivers. Such a characterization was previously unknown. Similar to Vatne's classification of the mutation class of $D_n$ quivers, our classification consists of different subtypes, where each subtype is a collection of blocks and subquivers in $\mathcal{M}^A$ or $\mathcal{M}^D$ glued along connecting vertices. However, there are many more subtypes compared to the type $D$ case, so we find it convenient to organize them into families: what we call *paired types*, quivers with one central cycle, and quivers with two central cycles.

We first describe the paired types and the role of explainability techniques in devising our characterization. Then we describe the quivers with one central cycle, which include six subtypes: V, Va, Vb, V', Va', Vb', and the quivers with two central cycles, which include two subtypes: VI and VI'. To show that this collection is the mutation class of $\tilde{D}_n$, we will adopt the strategy of [31] by showing that each type is mutation equivalent to (8) and then showing that this set of quivers is closed under quiver mutation.

## D.1  Paired types and PCA reductions of latent space embeddings

Applying Lemma C.1, we may choose an arbitrary orientation of the extended Dynkin diagram $\tilde{D}_n$. It will be convenient to begin with the orientation in (8), viewing it as two quivers $Q_1$ and $Q_2$ of type $D$ (7) connected at their roots by a connecting vertex $c$.

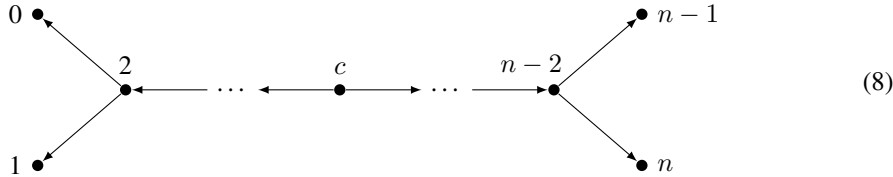

$$\tag{8}$$

From the orientation in (8) it is immediately clear that by mutating $Q_1$ and $Q_2$ independently without mutating $c$, we can obtain any pair of subtypes of type $D$. Because the placement of $c$ is arbitrary, we see that many type $\tilde{D}_n$ quivers can be described by two of the type $D$ subtypes characterized in Theorem 3.3 which share a type $A$ piece $\Gamma_c$. We will refer to such quivers as Types I-I, I-II, I-III, etc., and collectively as *paired types*. (See Fig. 13 in Appendix E for diagrams of all paired types.) It remains, then, to identify the quivers in this mutation class which not are of paired type.

While a human mathematician could conceivably discover the same characterization of $\tilde{D}$ quivers simply by beginning with (8) and exhaustively performing mutations, the mutation class of $\tilde{D}$ quivers

admits many diverse subtypes compared to classes $A$ or $D$. This increased complexity creates some difficulty (and perhaps more importantly, tedium) in examining examples manually. By taking advantage of machine learning, we are able to quickly organize examples into distinct families to examine.

Based on our strategy in Section 3, we plot PCA reductions of the latent space in Fig. 5. We can see that the quivers that do not correspond to paired subtypes, colored as "Other", separate clearly into two clusters in layer 3. By isolating these quivers and performing $k$-means clustering with $k = 2$, the model guides our characterization of the remaining class $\tilde{D}$ subtypes. Fig. 6 shows examples from each cluster. (More examples are given in Appendix E.) Examining the quivers in each cluster suggests that the remaining subtypes can be separated by the number of Type IV-like central cycles. We begin with quivers that have one central cycle.

## D.2 One central cycle

**Types V, Va, Vb.** Type V quivers resemble Type IV quivers of class $D$, but one edge in the central cycle is part of a $B_{\mathrm{IV}}$ block. In the diagram below, this is the edge $\alpha : a \to b$. Note that $d$ and $d'$ are dead ends in $B_{\mathrm{IV}}$. That is, no larger subquiver may be attached to $d$ and $d'$. Mutating Type V at $d$ produces the subtype **Type Va**, which is similar, but the block $B_{\mathrm{IV}}$ is replaced with an oriented 4-cycle, as shown below. Mutating type Va at $d'$ produces the subtype **Type Vb**, which is similar to (9) except the block $B_{\mathrm{IV}}$ is reversed. Mutating again at $d$ creates a quiver isomorphic to Type Va by swapping $d$ and $d'$, and finally mutating once more at $d'$ results in Type V again. Notice that the choice of $d$ and $d'$ is arbitrary.

**Type V.**      **Type Va.**      **Type Vb.**

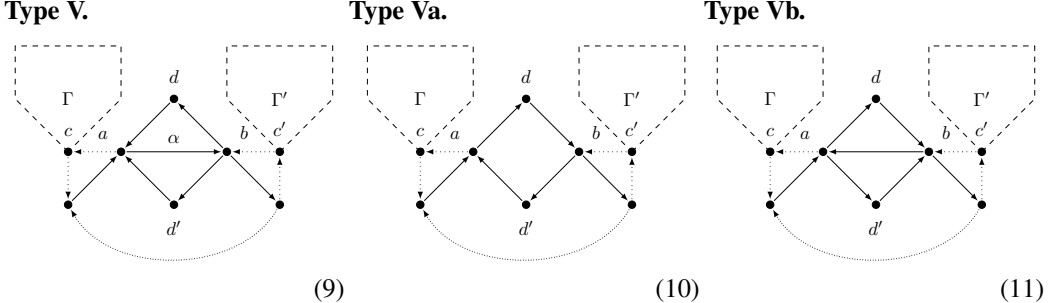

$$(9) \qquad\qquad (10) \qquad\qquad (11)$$

**Types V', Va', Vb'.** The rest of this cluster consists of the following types, which we call V', Va', and Vb' as they are related to each other by an analogous sequence of mutations. That is, starting from V', performing the sequence of mutations $\mu_d, \mu_{d'}, \mu_d, \mu_{d'}$ yields Types Va', Vb', Va', V', in that order. Moreover, $\mu_c$ converts each of Type V', Va', Vb' into a corresponding Type V, Va, or Vb quiver, respectively. The Type V' to Type V case is shown in Fig. 11.

**Type V'**      **Type Va'**      **Type Vb'**

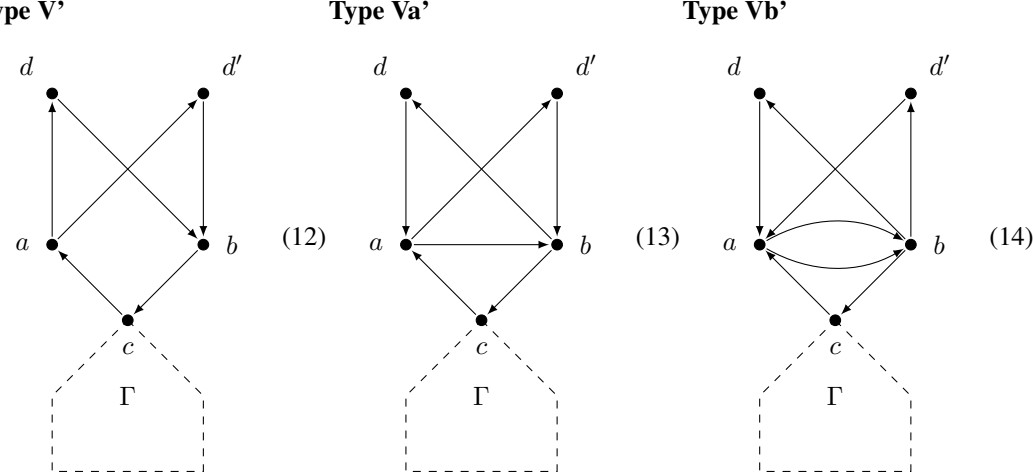

$$(12) \qquad\qquad (13) \qquad\qquad (14)$$

To see that these types are mutation equivalent to (8), it suffices to show that Type V is mutation equivalent to one of the paired types, which we show in Lemma D.1.

**Lemma D.1.** *Type V quivers* (9) *are mutation equivalent to* (8).

*Proof.* We mutate at vertex $a$. There are several cases. Recall from Theorem 3.3 that a *spike* refers to an oriented triangle on the central cycle.

If the central cycle is of length $> 3$, there are two subcases:

(a) If there is a spike at vertex $c$, then the resulting quiver is of Type II-IV, where the vertices $c$ and $a$ are playing the roles of $c$ and $c''$ in Figure 13, respectively, and $b$ plays the role of $c''$.

(b) Otherwise, the resulting quiver is of Type I-IV, where $d$ and $d'$ are the pair of dead ends.

If the central cycle is length 3, say a triangle $a \to b \to v \to a$, then there are four subcases, depending on the presence of spikes on the central cycle:

(a) If the central cycle has no additional spikes, then $\mu_a$ yields a Type I-I quiver.

(b) If $a$ is part of a spike but $b$ is not, then the result is a Type I-II quiver, where $b$ and $v$ are the pair of dead ends on the Type I side and $d, d'$ are the dead ends in the $B_{\text{IV}}$ block in the Type II side.

(c) If $a$ is not part of a spike but $b$ is then the result is Type I-III, where $d$ and $d'$ are the pair of dead ends in the Type I side.

(d) If both $a$ and $b$ are parts of spikes, then the result is Type II-III with dead ends $d, d'$ in the $B_{\text{IV}}$ block in the Type II side.

$\square$

**Corollary D.2.** *Quivers of Types Va (10) and Vb (11) are mutation equivalent to (8).*

**Corollary D.3.** *Type V' quivers (12) are mutation equivalent to (8).*

**Corollary D.4.** *Quivers of Types Va' (13) and Vb' (14) are mutation equivalent to (8).*

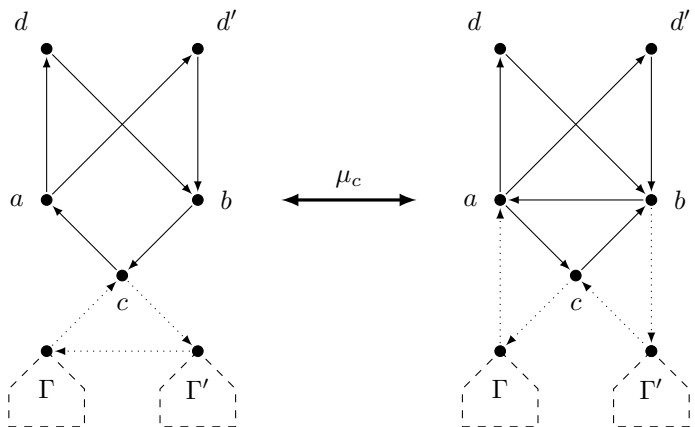

Figure 11: Performing $\mu_c$ to convert between a quiver of Type V' (left) and Type V (right). Note that in the Type V quiver, the central cycle is $a \to c \to b \to a$, so the vertex $c$ is not a connecting vertex as it is in 9.

This completes the description of new types with one central cycle.

### D.3  Two central cycles

The other cluster, consisting of quivers with two central cycles, consists of the following:

**Type VI.** Quivers of Type VI consist of two Type IV quivers which share one vertex $c$ among both central cycles, and are further joined by two edges that create oriented triangles for which $c$ is a vertex. These oriented triangles can be seen as shared spikes among both central cycles. In (15), we color one central cycle blue and one red for clarity. The central cycles may be any length $\geq 3$.

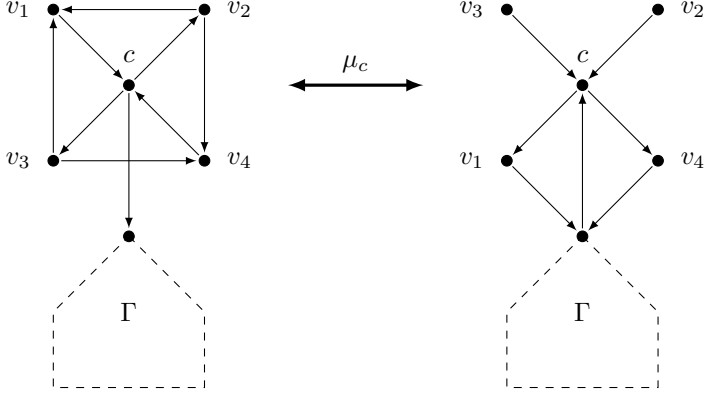

Figure 12: Performing $\mu_c$ on a quiver of Type VI' (left).

**Type VI'.** In Type VI, the shared connecting vertex $c$ is not allowed to be a connecting vertex for a larger subquiver of Type $A$ in general. However, there is one exception to this when both central cycles are triangles and have no additional spikes. The result is a block $B_V$ whose outlet is a connecting vertex for a type $A$ subquiver $\Gamma$. We refer to this as Type VI'. Notice that if the quiver has only five vertices, then $\Gamma$ is only one vertex, in which case there is no difference between Types VI and VI'.

**Type VI.**

**Type VI'.**

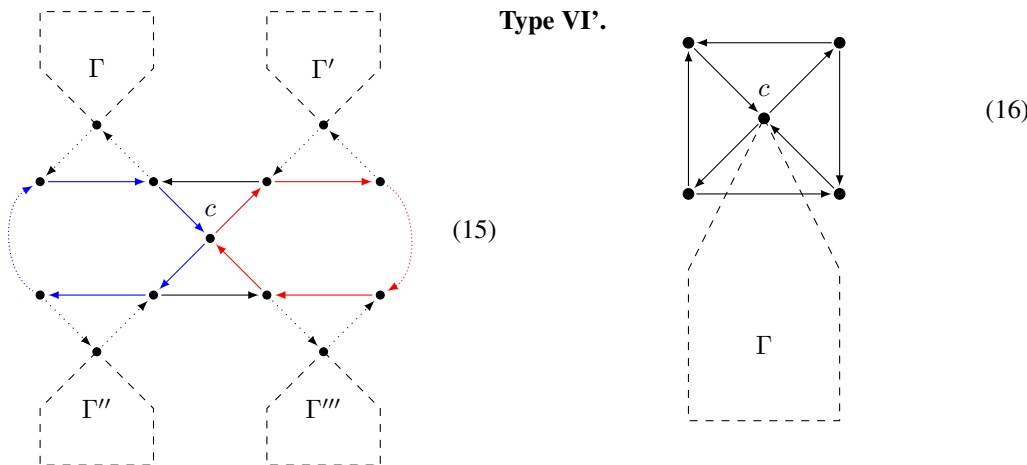

$$(15)$$

$$(16)$$

Again, to show that these are mutation equivalent to (8), it suffices to reduce to the paired types.

**Lemma D.5.** *Type VI quivers* (15) *are mutation equivalent to* (8).

*Proof.* If both central cycles are of length $> 3$, then mutating at vertex $c$ results in a quiver Type IV-IV. If a central cycle is of length 3, then $\mu_c$ turns that central cycle into a subquiver of Type I or Type III, depending on whether or not that central cycle does not have or does have a third spike, respectively. $\qquad\square$

**Lemma D.6.** *Type VI' quivers* (16) *are mutation equivalent to* (8).

*Proof.* By Lemma C.3, we may mutate so that $c$ has in-degree 0 and out-degree 1 in $\Gamma$. Then mutating at vertex $c$ results in a quiver of Type I-II (cf. Fig. 12). $\qquad\square$

## D.4 Proof of Theorem 3.1

*Proof.* From the preceding lemmas we know that these types are mutation equivalent to the quiver in (8), so we need only prove that these types are exhaustive by showing that $\mathcal{M}_n^{\tilde{D}}$ is closed under quiver mutation.

We will begin with the paired types. Suppose that $Q \in \mathcal{M}_n^{\tilde{D}}$ is the union of two quivers $Q_1, Q_2 \in \mathcal{M}^D$ whose intersection $\Gamma_c$ is in $\mathcal{M}^A$. If $\Gamma_c \in \mathcal{M}_k^A$ for $k > 1$, then any mutation can affect the type of at most one of $Q_1$ or $Q_2$ and hence by Theorem 3.3 results in a quiver of (possibly different) paired type. Thus in what follows we assume that $\Gamma_c$ is a single vertex $c$. Moreover, we need only consider mutating at $c$, since a mutation anywhere else can only convert $Q$ from one paired type to another.

In the casework below, when a type V quiver has a central cycle of length 3 and only 7 edges, we will refer to it as *minimal type V*. Similarly, a minimal type VI quiver is a type VI quiver where both central cycles are length 3.

**Type I-I.** Because we assume $\Gamma_c$ is a single vertex $c$, the underlying graph of this quiver is the star graph on 5 vertices, rooted at $c$. Mutating at $c$ depends on the number of arrows to and from $c$. If $c$ has indegree 4 or outdegree 4, then $\mu_c$ simply reverses every arrow. If $c$ has indegree 3 or outdegree 3, then $\mu_c$ produces a minimal Type V quiver. If $c$ has indegree 2 and outdegree 2, then $\mu_c$ produces Type VI (or VI', since they are the same when there are only 5 vertices).

**Type I-II.** Let $c'$ denote the connecting vertex opposite $c$ in the block $B_{IV}$ in the Type II subquiver, and $a, b$ denote the endpoints of the Type I arrows. Then if $Q$ contains the paths $a \to c \to c'$ and $b \to c \to c'$ or $c' \to c \to a$ and $c' \to c \to b$, then $\mu_c$ yields another Type I-II quiver. If $Q$ has $c \to c'$ with $c \to a$ and $c \to b$, or if $Q$ has $c' \to c$ with $a \to c$ and $b \to c$ the result is Type VI'. If $Q$ has $a \to c$ and $c \to b$ (or vice versa) then the result is Type V (regardless of the orientation of the $B_{IV}$ block.

**Type I-III.** If the Type I arrows are of the same orientation with respect to $c$, then $\mu_c$ results in a quiver of Type V. Otherwise, the result is Type VI.

**Type I-IV.** If the Type I arrows are of the same orientation with respect to $c$, then $\mu_c$ results in a quiver of Type V. Otherwise, the result is Type VI.

**Type II-II.** Let $c'$, $c''$ denote the connecting vertices opposite $c$ in the $B_{IV}$ blocks in the Type II subquivers $Q_1$ and $Q_2$, respectively. Then if the arrows are oriented $c' \to c \to c''$ (or the reverse) then $\mu_c$ yields a quiver of Type VI' where the connecting vertex $c$ is glues the $B_V$ block to an oriented triangle. Otherwise $\mu_c$ yields another Type II-II quiver.

**Type II-III.** Mutating at $c$ yields a Type V quiver (regardless of the $c - c'$ orientation).

**Type II-IV.** Mutating at $c$ yields a Type V quiver (regardless of the $c - c'$ orientation).

**Type III-III.** Mutating at $c$ yields a Type VI quiver with two central cycles of length 3.

**Type III-IV.** Mutating at $c$ yields a Type VI quiver.

**Type IV-IV.** Mutating at $c$ yields a Type VI quiver.

Having finished the paired types, we turn our attention to our newly identified types.

**Type V.** In the proof of Lemma D.1, we showed that mutating at $a$ stays in $\mathcal{M}^{\tilde{D}}$, producing a quiver of paired type in all cases. In Appendix D.2, we showed that mutating at $d$ (and $d'$ by symmetry) produces a Type Va quiver. If we mutate at $b$, we bifurcate into the same cases as when mutating at $a$ in Lemma D.1, and in fact obtain the same types. Now, if the central cycle is of length $> 3$, then we are done, as mutating anywhere along the central cycle will simply shrink the central cycle by 1, leaving us with another quiver of Type V. However, suppose the central cycle is of length 3, given by $a \xrightarrow{\alpha} b \to v \to a$. Then we must consider $\mu_v$, which yields Type V'. (In this manner, Type V' can be seen as the result of shrinking the central cycle to length 2, which we then remove because digons are prohibited.) Here the subquiver $\Gamma$ is a single vertex if there are no additional spikes, a single directed edge if there is one spike, and an oriented triangle if there are two spikes.

**Type Va.** We know from Appendix D.2 that $\mu_d$ and $\mu_{d'}$ yield Types V and Vb, respectively. Continuing, $\mu_a$ and $\mu_b$ both yield Type VI. If the central cycle (sans $\alpha$) is length $> 3$, then we are

done. Otherwise, we consider the case where we have a vertex $v$ with $b \to v \to a$ and compute $\mu_v$, which we see yields Type Va'.

**Type Vb.** From Appendix D.2 we see that $\mu_d$ and $\mu'_d$ yield Type Va. Mutating at $a$ or $b$ yields Type Vb again, simply moving the reversed arrow around the central cycle with the associated $d, d'$. Finally, we are done unless the central cycle is an unoriented triangle $a \leftarrow b \to v \to a$, in which case we must consider $\mu_v$, which yields Type Vb'.

**Type V'.** We know the mutations $\mu_d$ and $\mu_{d'}$ yield Type Va'. The mutations $\mu_a$ and $\mu_b$ yield Type VI'. Finally, we know that mutating at $c$ yields Type V from Corollary D.3 (see Fig. 11).

**Type Va'.** As we have seen, $\mu_d$ yields Type V' and $\mu_{d'}$ yields Type Vb'. The mutations $\mu_a$ and $\mu_b$ both result in Type Va' by cyclically permuting $(a, b, d)$ forwards and backwards, respectively. Finally, $\mu_c$ yields Type Va.

**Type Vb'.** The mutations $\mu_d$ and $\mu_{d'}$ yield Type Va'. Mutating at $a$ or $b$ produces an automorphism which swaps $a$ and $d$ with $b$ and $d'$, respectively, so $\mu_a$ and $\mu_b$ yield Type Vb' again. Finally, mutating at $c$ yields Type Vb.

**Type VI.** From Lemma D.5 we know that $\mu_c$ yields a paired type. Call the two central cycles $C_1$ and $C_2$. If we mutate at any vertex which is not adjacent to $c$, the result is still Type VI, as the mutation affects the relevant cycle $C_1$ or $C_2$ as a Type IV subquiver, and cannot break $C_1$ or $C_2$. Suppose then that we mutate at a vertex $v$ which is adjacent to $c$ and suppose, without loss of generality, that $v \in C_1$. Then $\mu_v$ simply moves $v$ from $C_1$ to $C_2$, resulting in Type VI, unless $C_1$ is a triangle, in which case the result is Type Va.

**Type VI'.** Since $c$ is a connecting vertex in $\Gamma$, it has degree at most 2. If $c$ has degree 1 in $\Gamma$, $\mu_c$ yields Type I-II, and if $c$ has degree 2 in $\Gamma$, $\mu_c$ yields Type II-II. Any other mutation in $\Gamma$ cannot change the type, so it remains only to check the vertices adjacent to $c$. Mutating at any results in Type V'.

Thus we have shown that $\mathcal{M}^{\tilde{D}}$ is closed under quiver mutation. This completes the proof. $\qquad\square$

# E   Additional figures

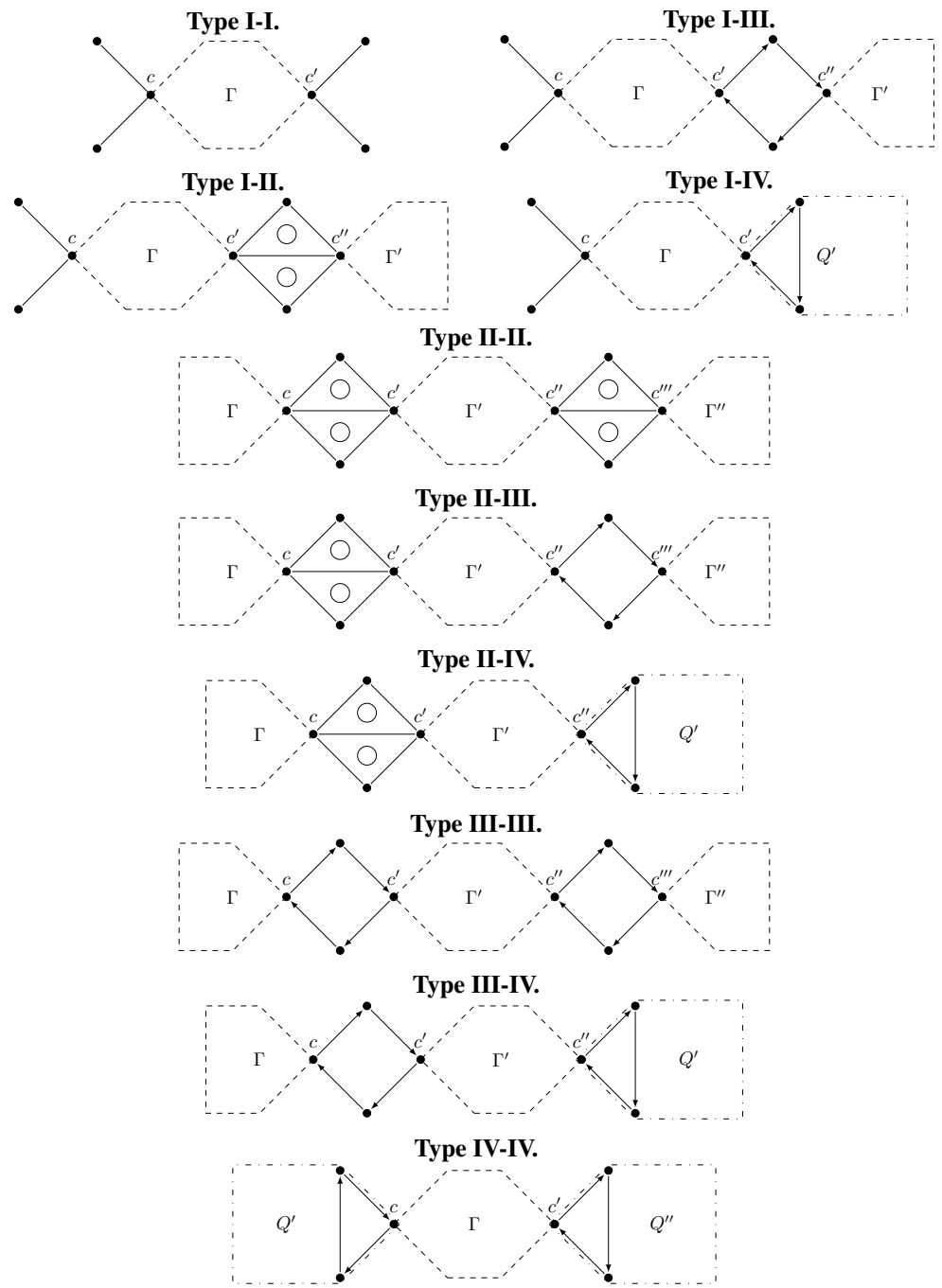

Figure 13: All paired types. Unoriented edges may have any orientation. Circles indicate oriented cycles. Here $\Gamma$, $\Gamma'$, $\Gamma''$ are subquivers of Type $A$, and $Q'$ is a subquiver of Type $D$-IV for which $c, c'$, or $c''$ is part of a spike. Notice that we may have $\Gamma \in \mathcal{M}_1^A$ with $c = c'$, but $Q'$ and $Q''$ must contain at least two edges in addition to the ones shown.

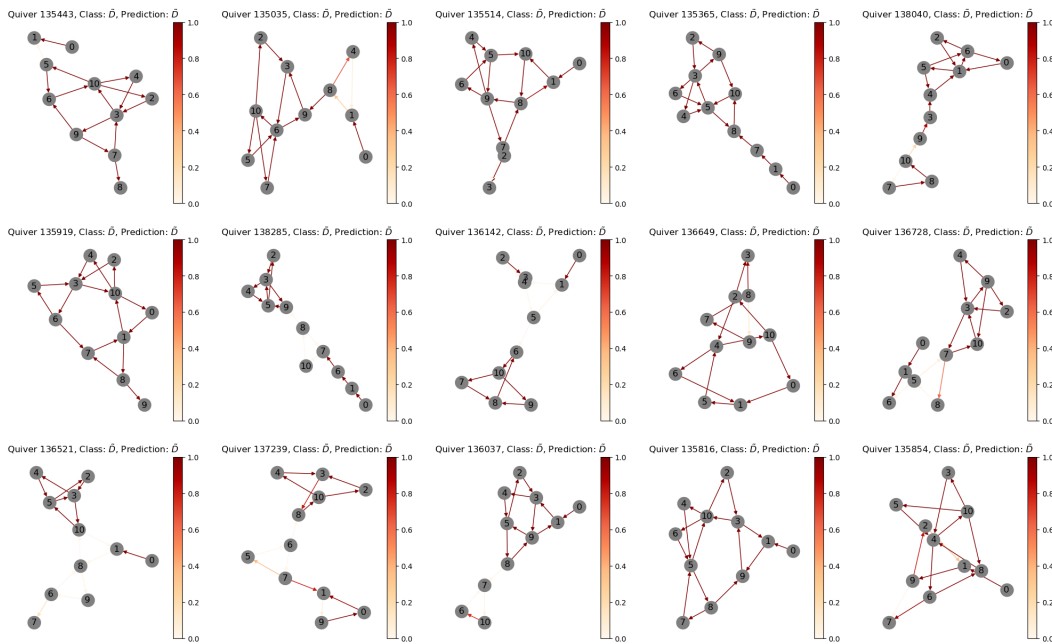

Figure 14: Randomly selected quivers from the orange (left) cluster in Fig. 6, which consists of quivers of Types V, Va, Vb, V', Va', Vb'.

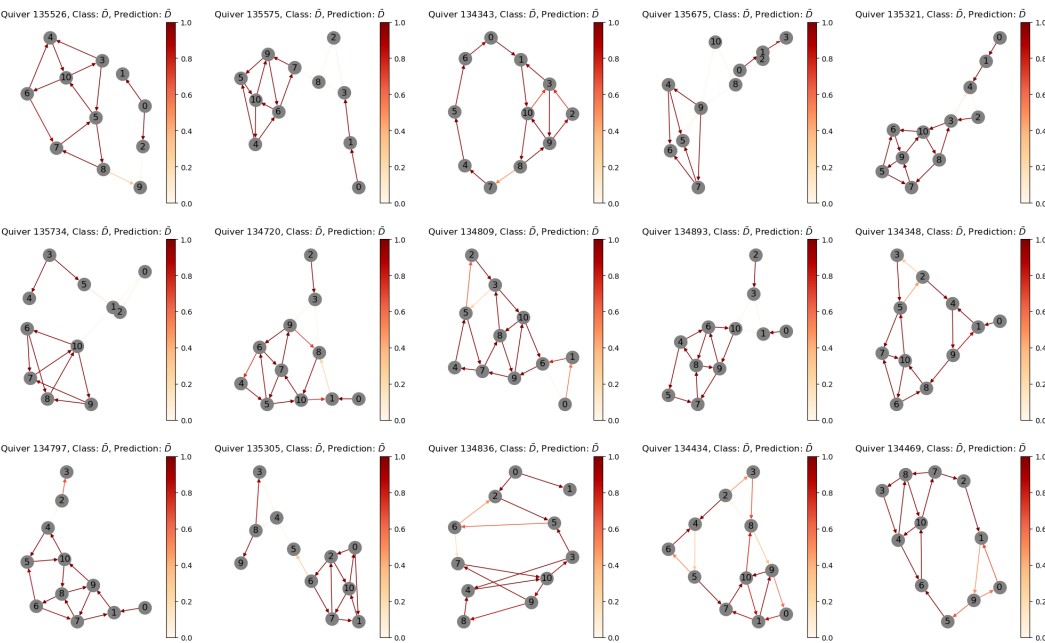

Figure 15: Randomly selected quivers from the blue (right) cluster in Fig. 6, which consists of quivers of Types VI and VI'.

