# OpenReview forum: "Machines and Mathematical Mutations: Using GNNs to Characterize Quiver Mutation Classes"
_NeurIPS.cc/2024/Workshop/MATH-AI — MATH-AI 24_

### Official Review · Reviewer_XnBa · 2024-10-02
**An interesting application of deep learning to mathematical discovery**

**Rating:** 9
**Confidence:** 5

**Review:**

The authors train graph neural networks to predict the mutation classes of quivers, directed multigraphs without 2-loops or cycles. Models trained on quivers with 6 to 10 nodes, can predict the classes of quivers wit 11 nodes with very high accuracy.

The authors then study the latent space embedding of trained models, and show that they reflect known typologies of quiver classes, suggesting that the models have learned "something mathematical".

This is a very interesting application of machine learning to a current research problem in graph theory. It would deserve exposure in the Math&AI workshop. Oral?

---

### Official Review · Reviewer_wBpZ · 2024-10-07
**The paper introduces a GNN-based approach to assess quiver mutation equivalence. I've provided some potential minor improvements below.**

**Rating:** 6
**Confidence:** 3

**Review:**

This paper leverages DirGINE to predict quiver mutation equivalence. The visualizations presented in Figure 3-5 are interesting and showcase the practical value of this approach. I would suggest adding some quantitative results and discuss accordingly in the main text, which could help readers gain a better sense of the proposed method.

---

### Official Review · Reviewer_cqAP · 2024-10-07

**Rating:** 8
**Confidence:** 4

**Review:**

The paper uses graph neural networks to study quiver mutation problem. The authors provide extensive theoretical analysis and case study, conduct experiments and utilize techniques in explainability of GNNs to interpret the results. The paper could benefit from more investigation with more expressive GNN variants and more state-of-the-art GNN explainability techniques.

---

### Decision · Program_Chairs · 2024-10-09

Accept